# HRBP: Hardware-friendly Regrouping towards Block-based Pruning for Sparse CNN Training

Haoyu Ma[1,*] Chengming Zhang[2]*, Lizhi Xiang[3]*, Xiaolong Ma[4],
Geng Yuan[5], Wenkai Zhang[1], Shiwei Liu[6,7,8], Tianlong Chen[9,10,11],
Dingwen Tao[2], Yanzhi Wang[12], Zhangyang Wang[6], Xiaohui Xie[1]
[1]University of California, Irvine, [2]Indiana University Bloomington, [3]University of Utah,
[4]Clemson University, [5]University of Georgia, [6]University of Texas at Austin,
[7]Eindhoven University of Technology, [8]University of Oxford, [9]MIT, [10]Harvard University,
[11]The University of North Carolina at Chapel Hill, [12]Northeastern University
{haoyum3,wenkaiz1,xhx}@uci.edu, {czh5,ditao}@iu.edu, lxiang@cs.utah.edu,
xiaolom@clemson.edu, geng.yuan@uga.edu, s.liu3@tue.nl, tianlong@mit.edu,
yanz.wang@northeastern.edu, atlaswang@utexas.edu

Pruning at initialization and training a sparse network from scratch (sparse training) become increasingly popular. However, most sparse training literature addresses only the unstructured sparsity, which in practice brings little benefit to the training acceleration on GPU due to the irregularity of non-zero weights. In this paper, we work on sparse training with fine-grained structured sparsity, by extracting a few dense blocks from unstructured sparse weights. For Convolutional Neural networks (CNN), however, the extracted dense blocks will be broken in backpropagation due to the shape transformation of convolution filters implemented by GEMM. Thus, previous block-wise pruning methods can only be used to accelerate the forward pass of sparse CNN training. To this end, we propose Hardware-friendly Regrouping towards Block-based Pruning (HRBP), where the grouping is conducted on the kernel-wise mask. With HRBP, extracted dense blocks are preserved in backpropagation. Extensive experiments on CIFAR-10, CIFAR-100, and ImageNet demonstrate that HRBP can almost match the accuracy of unstructured sparse training methods while achieving a huge acceleration on hardware. Code is available at `https://github.com/HowieMa/HRBP-pruning`.

## 1. Introduction

Convolutional Neural Networks (CNN) have accomplished enormous progress on many computer vision tasks, such as classification, detection, and segmentation. However, most successful models are overparameterized and computationally extensive. The excessive computation usually requires tedious training and makes it difficult to deploy into real applications. Network pruning [1–4], which removes unnecessary weights from the heavy dense model, stands as one of the most effective methods to compress a heavy model into a lightweight counterpart while maintaining its accuracy.

Conventional network pruning usually starts from pre-training a dense DNN model, and then utilizes specific heuristics to prune the model parameters to obtain a sparse DNN model [2–4]. However, this paradigm is still inefficient as it needs to train the dense model first. The recent Lottery Ticket Hypothesis (LTH) [5] suggests that a sparse network can be trained from scratch (sparse training) to the same accuracy as its original dense model. Consequently, the tedious dense training is unnecessary. During the training process, The sparse structure (sparse mask) can either be static [6–8] or dynamic [9–11]. Most sparse training methods [6–11] explore unstructured sparsity only, where zero weights distribute irregularly. Although unstructured sparsity can maintain accuracy at a high sparsity ratio[8], it brings little training time reduction on modern hardware because the irregular mask leads to poor data locality and low parallelism [12–14]. An alternative approach,

---

[*]These authors contributed equally

First Conference on Parsimony and Learning (CPAL 2024).

structured sparsity [12, 15], where the entire filter or channel is pruned, is more hardware-friendly and computationally efficient for sparse training [16, 17]. However, structured sparsity usually suffers a notable accuracy drop when the pruning rate increases.

Recently, there has been a surge in popularity for *fine-grained structured sparsity*, which includes pattern-based [18–20] and block-based [21] approaches, due to its ability to retain the benefits of unstructured and structured sparsity. A few works have explored sparse training with fine-grained structured sparsity based on the N: M sparsity [20, 22, 23], in which only $N$ weights are non-zero for every continuous $M$ weight. The N: M transposable mask [24] further ensures that both the weight matrix and its transpose follow the same sparsity pattern, which help accelerate both forward and backward passes and the recent Bi-Mask [23] involves two different masks during the sparse training. However, these methods require specialized hardware, *i.e.*, the sparse tensor cores [25]. Besides, the transposable mask cannot be directly applied to arbitrary CNN training. As shown in Fig. 1, the general matrix multiplication (GEMM) implementation of CNN [26] on hardware calculates the input gradient by first rotating each kernel and then performing a kernel-wise transpose, instead of a straightforward transpose operation (See Sec. 3.1 for more detail). As a result, it is possible that the transposable masks may not yield the expected speed-up during the backward pass of the CNN. Meanwhile, the regrouping method [21, 27] extracts dense blocks by grouping unstructured sparse weights. This approach has been shown to be effective in accelerating computations on modern hardware. However, as shown in Fig. 2, the dense blocks that are extracted during the forward pass of CNN are typically not preserved in the backward pass, which limits the ability to accelerate backpropagation.

In this paper, we aim to accelerate the sparse training of CNNs using fine-grained structured sparsity through regrouping during both the forward and backward passes. Specifically, we propose Hardware-friendly Regrouping towards Block-wise Pruning (HRBP), which performs the regrouping algorithm on the kernel-wise mask. Thus, it has the ability to maintain the same dense blocks at both forward and backward passes of CNN. Besides, HRBP extracts exclusive sparse pattern for each group to achieve a more fine-grained sparsity. Additionally, all of the blocks extracted by HRBP are of the same shape, which can help alleviate issues related to unbalanced workloads on many-core GPUs [28]. We show that that static sparse training using HRBP can achieve nearly the same accuracy as unstructured sparse training methods, while offering significant training acceleration. Moreover, we propose a block-wise updating algorithm to facilitate the application of HRBP in dynamic sparse training. Our main contributions are summarized as follows:

- Our analysis of CNN's forward and backward pass using GEMM reveals that current fine-grained structured pruning methods do not ensure accelerated backward propagation.

- We propose a novel Hardware-friendly Regrouping Block-wise Pruning (HRBP), which accelerates both the forward and backward passes of CNN training by extracting dense blocks from non-zero weights while preserving spatial regularity.

- Extensive experiments on CIFAR and ImageNet demonstrate that HRBP enables better accuracy and hardware acceleration trade-offs in both static and dynamic sparse training.

## 2. Related work

### 2.1. Network Pruning

Pruning aims to compress overparameterized networks into lightweight ones. Based on the distribution of zero weights, it can be divided into three types: 1) *Unstructured sparsity*, where zero weights are distributed at arbitrary locations based on the importance score of each weight. The score can be obtained from magnitude [1–3, 29, 30], gradient [31, 32] or Hessian [1]. Unstructured pruning can achieve high sparsity ratios while maintaining accuracy, but it is difficult to speed up on hardware due to its irregularity. 2) *Structured sparsity*, where the weights of entire channels are pruned. Earlier works [4, 12, 14, 15, 17] adopt mathematics-oriented regularization-based algorithms to generate sparsity. Other works such as HRank [33], SCOP [34], and DMCP [35] use complicated rules to generate the sparsity distribution in the channel level. As the sparse model

preserves the spatial regularity, the pruned convolution layers can be transformed to a full matrix multiplication with reduced matrix size and accelerate computation on the hardware level. However, structured pruning suffers from significant accuracy loss as one entire activation map can be zero. 3) *Fine-grained structured sparsity*, which includes pattern-based pruning[18–20] and block-based pruning [21]. In pattern-based pruning [18], the weights can only be pruned to one of several pre-defined sparse patterns. Thus, it is limited to a few pruning options and pruning ratios. On the contrary, the block-based pruning [21] is more flexible as it directly extracts dense blocks from any sparse weights. Thus, one single tedious sparse matrix multiplication can be achieved by carrying out multiple small dense matrix multiplications with GeMM.

## 2.2. Sparse Training

Sparse training aims to train a pruned sparse network from scratch. Based on the mask updating schemes, it is usually divided into two categories: 1) *Static sparse training* (*SST*), where the sparse mask is obtained at the early stage of training and is fixed during the course of training. Previous works obtain the sparse mask by random pruning or utilizing some saliency criteria, such as the gradients of the training loss [6], gradient flow in GraSP [7], synaptic strengths in SynFlow [8], Fisher information [36], etc. 2) *Dynamic sparse training* (*DST*), which starts from a random sparse network. After optimizing several iterations, it prunes a portion of weights based on the pruning criterion and grows new connections according to the grow criterion. Then the new sparse network is trained until the next update. Specifically, SET [9] updates sparse masks by pruning weights that have the least magnitude and grow back the same amount of inactivated weights in a random fashion. RigL [10] proposes to update sparse masks by magnitude-based pruning and grow back inactivated weights by their gradients. DSR [37] and STR [38] design a dynamic reparameterization method that allows weights to be re-distributed across layers by providing a global sparsity allocation dynamics. DeepR [39] combines dynamic sparse parameterization with stochastic parameter updates for training, but it primarily targets small and shallow fully-connected networks. However, most works in this area focus on unstructured sparsity only.

## 3. Preliminaries

### 3.1. Convolution Operation and Its Implementation

A 2D convolutional layer's weights are defined as $\mathbf{K} \in \mathbb{R}^{C_O \times C_I \times K_h \times K_w}$, where $C_O$, $C_I$, $K_h$ and $K_w$ are the number of output channels, the number of input channels, kernel height, and kernel width, respectively. During the convolution operation, each filter $\mathbf{K}_c$ slides over the input feature map $\mathbf{I} \in \mathbb{R}^{C_I \times H_I \times W_I}$, computing a weighted sum of the input values, resulting in one activation map $\mathbf{O}_c \in \mathbb{R}^{H_O \times W_O}$. The $C_O$ filters perform $C_O$ times of convolution operations, generating the output map $\mathbf{O} \in \mathbb{R}^{C_O \times H_O \times W_O}$. See Appendix A for a summary of all notations in this paper.

**Forward pass with GEMM.** On hardware, convolution operations are usually implemented using general matrix-matrix multiplication (GEMM) [26]. The tensor is laid out in memory in either the `NCHW` or `NHWC` format (See Appendix for more details). We take the `NCHW` format as an example. As shown in Fig. 1 (a), for the input $\mathbf{I}$, the `im2col(·)` operation flattens each convolution window of the input and stacks them as columns in a matrix. Thus, the 2D input feature map $\mathbf{I}$ is unrolled into an input matrix $\mathbf{X} = \texttt{im2col}(\mathbf{I}) \in \mathbb{R}^{(C_I K_h K_w) \times (H_O W_O)}$. Meanwhile, $\mathbf{K}$ is reshaped and stored in the weights matrix $\mathbf{W} \in \mathbb{R}^{C_O \times (C_I K_h K_w)}$. The forward pass is calculated by $\mathbf{Y} = \mathbf{WX} \in \mathbb{R}^{C_O \times (H_O W_O)}$, and the 2D output map $\mathbf{O}$ is obtained by reshaping $\mathbf{Y}$.

**Backward pass with GEMM.** Given the gradients of the 2D output map $\mathbf{dO} \in \mathbb{R}^{C_O \times H_O \times W_O}$, the backpropagation involves two matrix multiplications. 1) Calculate the gradients w.r.t. the filters $\mathbf{dK}$, which is implemented by $\mathbf{dW} = \mathbf{dY} \cdot \mathbf{X}^T$ following $\mathbf{dK} = \texttt{reshape}(\mathbf{dW})$. 2) Calculate the gradients w.r.t the input $\mathbf{dI}$, which can be obtained by a full convolution between the kernel $\mathbf{K}$ and $\mathbf{dO}$ [40]. In detail, as in Fig.1 (b), we conduct padding and `im2col()` operation on $\mathbf{dO}$, and obtain $\mathbf{dY} = \texttt{im2col}(\mathbf{dO}) \in \mathbb{R}^{(C_O K_h K_w) \times (H_I W_I)}$. Meanwhile, we flip each kernel first vertically and then horizontally (*i.e.*, $180°$ rotation) and perform the kernel-wise transpose to get the new kernel layout $\mathbf{K}'$. Then we reshape $\mathbf{K}'$ to matrix $\mathbf{W}' \in \mathbb{R}^{C_I \times (C_O K_h K_w)}$. Thus, the gradient is calculated by $\mathbf{dX} = \mathbf{W}'\mathbf{dY}$ and finally reshaped to obtain $\mathbf{dI}$.

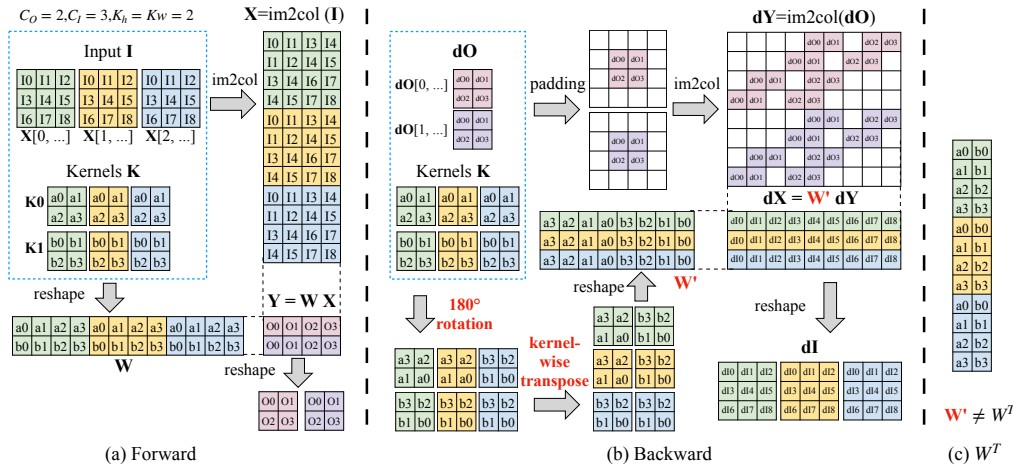

(a) Forward

(b) Backward

(c) $W^T$

Figure 1: Implementation of forward and backward pass of convolution operation with GEMM in `NCHW` layout. Different color represents different channels. In the forward pass, the kernels **K** are reshaped to matrix **W**. In the backward pass, each filter is rotated $180°$ firstly, then the kernel-wise transpose is conducted to obtain the new kernel layout **K′**. Then **K′** is reshaped to matrix **W′**, which is different from the transpose **$W^T$**.

**Discussion.** Previous works [24] calculate the gradient w.r.t the input by $\mathbf{dX} = \mathbf{W^T dY}$ for simplicity, where $\mathbf{W^T} \in \mathbb{R}^{(C_I K_h K_w) \times C_O}$, which is applicable for linear layers. However, this formulation cannot be generalized to CNN. As in Fig.1 (c), $\mathbf{W^T}$ is different from $\mathbf{W'}$ in CNN. The only case where they are the same is when $K_h = K_w = 1$, which reduces the CNN to a linear layer. Thus, the backward pass of CNN cannot be linearly simplified to $\mathbf{dX} = \mathbf{W^T dY}$ in general. Consequently, sparse patterns based on $\mathbf{W^T}$ [24] may not yield the expected acceleration in arbitrary CNN backpropagation.

## 3.2. Weight Regrouping on Unstructured Sparsity

Given a CNN $f_\theta(\cdot)$ with weights $\theta$, a sparse subnetwork is defined by $f_{\theta \odot \tilde{m}}(\cdot)$, where $\tilde{m} \in \{0, 1\}^{|\theta|}$ is the binary mask and $\odot$ is the element-wise product. In unstructured pruning, the zeros are unevenly distributed in $\tilde{m}$. Thus, the entire weight matrix still needs to be maintained as a dense network, making it difficult to reduce the computation of unstructured sparse weights on hardware. Even with the help of the dedicated sparse matrix representation technique such as CSR format [41], the unstructured sparsity still expects to have over $85\%$ sparsity ratio to acquire limited acceleration since the irregular weight distribution causes significant computation overhead due to poor data locality [27]. The recent weight regrouping (reorganization) [21] can accelerate the unstructured sparse weights on hardware by extracting multiple smaller dense blocks in a large sparse matrix, which can improve the throughput with GEMM.

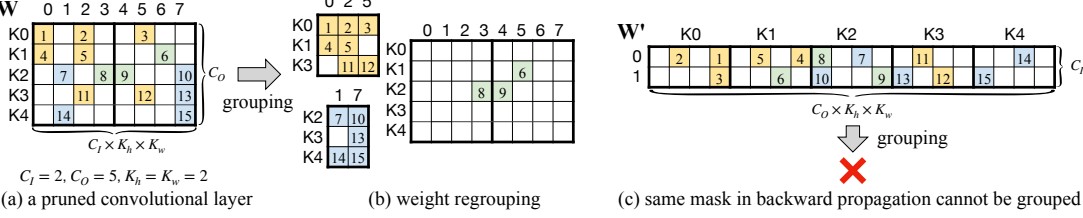

(a) a pruned convolutional layer

(b) weight regrouping

(c) same mask in backward propagation cannot be grouped

Figure 2: Example of weight regrouping on convolutional operation. Indexed cells are non-zeros. Different color represents different block groups. The extracted dense blocks in forward pass (b) cannot be kept in the backward pass (c).

**Implementation.** Denote the binary mask of the weights of a CNN layer as $m \in \{0, 1\}^{|\mathbf{W}|}$. The regrouping algorithm [21] finds similar rows and columns from the sparse weights matrix $\mathbf{W} \odot m$ and brings them together into several dense blocks. The process begins by clustering $C_O$ rows of $m$

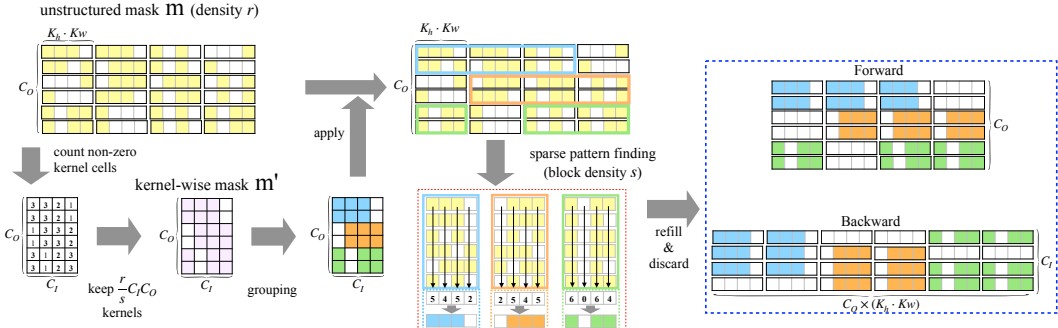

Figure 3: Illustration of HRBP. White cell means zero weights. Given the unstructured mask (matrix with yellow cell) with density $r$, HRBP first counts the number of non-zero cells of each kernel and extracts the kernel-wise mask $m'$ (matrix with lavender cell) with density $\frac{r}{s}$. The kernel-wise mask is then grouped into dense blocks with equal shapes, with $t = 3$ groups being marked with blue, orange, and green. Furthermore, exclusive sparse patterns of each block are extracted and applied to all kernels within the group.

into several groups based on the Jaccard similarity among non-zero columns. Each group is then further processed by selecting columns with the most non-zero weights from all $C_I K_h K_w$ columns, resulting in a single dense block for each group. For instance, in Fig. 2(a), filters $\mathbf{K}_0$, $\mathbf{K}_1$, and $\mathbf{K}_3$ can be grouped together, and columns with at least two non-zero weights (i.e., $0^{th}$, $2^{th}$, and $5^{th}$ columns) can be selected to form a dense block with an orange color, as illustrated in Fig. 2(b). The remaining sparse weights, shown in green, are usually discarded [16].

**Limitations.** However, the regrouping algorithm cannot be applied to the training of CNN directly. Firstly, the extracted dense blocks are fragmentary in backward pass due to the transformation from $\mathbf{W}$ to $\mathbf{W}'$, making the backward acceleration unfeasible. An illustration is shown in Fig. 2(c), where the cells of the dense block with orange color in $\mathbf{W} \odot m$ are scattered irregularly in $\mathbf{W}'$. Besides, the shapes of different dense blocks are arbitrary, which introduces imbalanced memory access and data locality. Thus, it makes the GPU suffer a great workload imbalance. Hence, a novel regrouping methodology is needed to expedite the sparse training of CNN in forward and backward passes.

# 4. Methodology

## 4.1. Hardware-friendly Regrouping for Block-wise Pruning (HRBP)

**Motivation.** The issue of fragmentary blocks in CNN emerges from the independent consideration of all $C_I K_h K_w$ columns, allowing a dense block to select arbitrary locations on the kernel across various different input channels. Ideally, the grouping algorithm should locate identical elements for different input channels within one block. Naturally, when $K_h = K_w = 1$, the issue of fragmentary blocks can be solved as the CNN reduces to a linear layer. Inspired by this property, we propose the HRBP, which extracts dense blocks that can be kept in CNN backpropagation. HRBP upholds two key properties: it conducts regrouping based on a kernel-wise mask and extracts exclusive sparse patterns within each dense block.

**Kernel-wise mask grouping.** HRBP performs regrouping on the kernel-wise mask $m' \in \{0,1\}^{C_O \times C_I}$, instead of the unstructured mask $m \in \{0,1\}^{C_O \times (C_I K_h K_w)}$ as utilized in [21]. Specifically, given mask $m$ with density ratio $r$, we first count the number of non-zero cells in each kernel based on $m$. For example, the top-left kernel contains 3 non-zero cells. We subsequently derive a kernel-wise mask denoted as $m'$ by retaining $\frac{r}{s}$ kernels with the highest non-zero cell counts. Here, $s$ is the block-wise density (with $s \geq r$ and $s \in \{i/(K_h * K_w)\}_{i=1}^{K_h * K_w}$), and a detailed illustration will be presented later. Next, we cluster the $C_O$ rows of $m'$ into $t$ groups of equal size based on the Jaccard similarity [21]. For each group, we choose $\frac{r}{s} \cdot C_I$ columns with the highest non-zero cell counts. For instance, rows 1 and 2 are clustered together as a group, depicted in blue, and the initial three columns from the left are kept. Finally, we extend the groupings obtained from $m'$ onto the original element-wise mask $m$, resulting in the creation of dense blocks. As shown in Fig. 3, the

same dense blocks are maintained during backpropagation with the kernel-wise mask, enabling the use of block-based sparse training in both forward and backward passes.

**Exclusive block sparse pattern.** Although the kernel-wise mask solves the backpropagation issue, it either retains or discards all cells within a single kernel. This may produce a large number of zero kernels, which can cause a significant reduction in accuracy [8]. Thus, we introduce the block sparsity to achieve more fine-grained and diverse level of sparsity within each kernel and reduce the number of zero kernels. Specifically, for each group, we count the number of non-zero weights across all kernels within the group, and choose $sK_hK_w$ cells with the highest number of non-zero weights as an exclusive sparse pattern for the dense block. For instance, in Fig. 3, the blue group contains six kernels, and each kernel has four cells. The number of non-zero weights for each cell is $5, 4, 5,$ and $2$, respectively. Thus, we select the 1st, 2nd, and 3rd cell as a sparse pattern and apply it to all kernels in the blue group, while select the 2nd, 3rd, and 4st cell for the orange group. In this way, we obtain $t$ dense blocks with an identical shape of $\frac{C_o}{t} \times (rC_IK_hK_w)$. See Appendix D for a detailed pseudocode and Appendix E for hardware implementation.

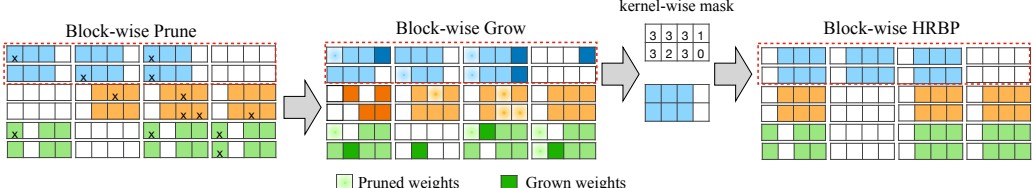

Figure 4: Block-wise updating for Dynamic Sparse Training. Take the block with blue color as an example, for each row, we prune 2 weights (crossed cells) based on magnitude and grow back 2 inactivated weights (darker cells) based on gradients. We then rerun HRBP to identify a new dense block for each group by generating kernel-wise masks to determine the kept kernels and applying pattern findings within each group.

## 4.2. Sparse Training with HRBP

**Static Sparse Training with HRBP.** Using HRBP, we are able to extract dense blocks from any form of sparse weights, resulting in a practical acceleration of both forward and backward passes during sparse training. In static sparse training, the sparse mask $m$ is generated before training. To simplify the process, we by default apply HRBP to a random unstructured mask to obtain dense blocks. This is equivalent to randomly dividing the $C_O$ rows into $t$ equal groups and keeping $rC_I$ channels random for each group since the initial mask is random. We name this special random sparse pattern as *HRBP-based Random Mask*. The Erdős–Rényi-Kernel (ERK) approach [10] is used to determine the sparse ratio of each layer, which allocates higher sparsity to larger layers compared to smaller ones within a network.

**Dynamic Sparse Training with HRBP.** In SST, HRBP identifies exclusive sparse patterns within each group using initial random masks, which may not result in optimal connections. Thus, we can leverage DST to further enhance connectivity during the training time. We follow the widely-used update mechanism from RigL [10], which prunes weights based on their magnitude and grows weights based on their gradients. However, one drawback of the RigL approach is that it updates connections at arbitrary locations, which could potentially destroy the extracted dense blocks in HRBP. To address this issue, we propose *block-wise updating*, which preserves the groups of rows (filters) and only updates the column-wise masks within each group. Specifically, as illustrated in Fig. 4, we first prune $d\%$ weights based on their magnitude and subsequently grow back new weights by $d\%$ based on their gradient for each row. Then, for each group, we rerun the HRBP on the updated weights to extract one dense block.

# 5. Experiments

## 5.1. Experimental Setups

**Dataset & Networks.** We follow the settings in [7, 27] and conduct experiments on CIFAR-10, CIFAR-100 and ImageNet-1K [42]. By default, we apply the WideResNet-32-2, WideResNet-56-2, and VGG-19 [43] for CIFAR-10/100 and ResNet-50 [44] for ImageNet.

**Training.** For CIFAR-10/100, we use a batch size of $128$ and train networks with SGD optimizer for 160 epochs. The learning rate is set to $0.1$ initially and is decayed by a factor of $0.1$ at the 80th and 120th. Moreover, we run each experiment 3 times and report the mean value and standard derivation. For ImageNet, we adopt the Pytorch official implementation and train the networks for 100 epoch as [7]. The learning rate is $0.1$ initially and is decayed at 30-th, 60-th, and 90-th epoch with factor $0.1$. For sparse training, we set the number of dense blocks $t$ to $8$ [21] and the block density $s$ to $\frac{4}{9}$ for all $3 \times 3$ kernels. The minimum size of a block $B_1$ is set to 8 [21].

**Hardware.** We evaluate our method on NVIDIA Ampere A100 (108 SMs, 40GB). The versions of CUDA and cuDNN are 11.0.0 and 8.0.4, respectively. We adopt the GEMM-based convolution, i.e., `CUDNN_CONVOLUTION_FWD_ALGO_GEMM` [26] as the baseline implementations of sparse convolution operations. We report the overall training time acceleration rate, which is determined by comparing the end-to-end training time in practice of sparse networks with that of the dense models. This encompasses both the forward and backward computations on the hardware.

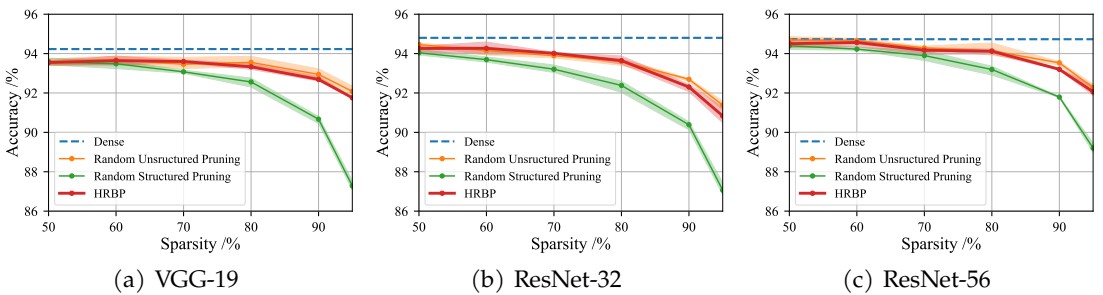

|     (a) VGG-19     |    (b) ResNet-32    |    (c) ResNet-56    |

Figure 5: The trade-off between sparsity and accuracy. All models are evaluated on CIFAR-10. Each line on the graph represents the mean value of three experimental runs, while the shaded area around it represents the variance among those three experiments.

## 5.2. Static Sparse Training with HRBP

Table 1: Comparison of static sparse training methods on CIFAR-10 and CIFAR-100. The number in brackets is the relative end-to-end training time speedup compared to a dense model.

| Network | ResNet-32 | | | | VGG-19 | | | |
|---|---|---|---|---|---|---|---|---|
| Dataset | CIFAR-10 | | CIFAR-100 | | CIFAR-10 | | CIFAR-100 | |
| Dense | 94.80 | | 74.64 | | 94.23 | | 74.16 | |
| Sparsity | 90% | 95% | 90% | 95% | 90% | 95% | 90% | 95% |
| RU | 92.81±0.19 (1.0x) | 91.38 ±0.04 | 69.48±0.21 (1.0x) | 67.03±0.66 (1.0x) | 92.91±0.10 (1.0x) | 91.91±0.13 (1.0x) | 70.39±0.43 (1.0x) | 68.63±0.40 (1.0x) |
| LTH [5] | 92.31 (1.0x) | 91.06 (1.0x) | 68.99 (1.0x) | 65.02 (1.0x) | 93.51 (1.0x) | 92.92 (1.0x) | 72.78 (1.0x) | 71.44 (1.0x) |
| SNIP [6] | 92.59 (1.0x) | 91.01 (1.0x) | 68.89 (1.0x) | 65.22 (1.0x) | 93.63 (1.0x) | 93.43 (1.0x) | 72.84 (1.0x) | 71.83 (1.0x) |
| GraSP [7] | 92.38 (1.0x) | 91.39 (1.0x) | 69.24 (1.0x) | 66.50 (1.0x) | 93.30 (1.0x) | 93.04 (1.0x) | 71.95 (1.0x) | 71.23 (1.0x) |
| RC | 90.27±0.24 (1.5x) | 87.19±0.48 (1.6x) | 62.71±0.22 (1.5x) | 56.33 ±0.37 (1.6x) | 90.84±0.38 (1.7x) | 87.12±0.18 (1.8x) | 59.61±0.52 (1.7x) | 49.31±1.17 (1.8x) |
| Grouping [21] | 91.63±0.11 (1.1x) | 90.77±0.07 (1.2x) | 66.97±0.18 (1.1x) | 64.35±0.45 (1.2x) | 92.81±0.25 (1.2x) | 91.86±0.24 (1.2x) | 70.52±0.36 (1.2x) | 68.60±0.08 (1.2x) |
| HRBP | 92.30±0.20 (1.4x) | 90.84±0.41 (1.6x) | 69.22±0.50 (1.4x) | 65.94±0.25 (1.6x) | 92.88±0.12 (1.4x) | 91.66±0.14 (1.9x) | 70.25±0.29 (1.4x) | 67.89±0.49 (1.9x) |

**Accuracy.** As we by default extract dense blocks with HRBP from random unstructured masks, our HRBP-based random mask belongs to the random pruning family. Consequently, we can make a fair comparison of our method with random unstructured pruning (RU) [45] and random channel-wise (structured) pruning (RC) with an ERK layer-wise ratio. We vary the sparsity ratio $1 - r$ from $50\%$ to $95\%$ to explore the trade-off between sparsity and the accuracy of different methods. Fig. 5 demonstrates that, across all sparsity levels, HRBP achieves comparable performance to random unstructured pruning techniques and significantly outperforms random channel-wise pruning. Moreover, we compare HRBP on static sparse training against several unstructured pruning techniques with well-defined criteria including SNIP [6] and GraSP [7] and LTH [5]. We also evaluate the vanilla regrouping method [21] as a baseline, even if it fails to accelerate the backpropagation. We are super interested in the accuracy at large sparsity ratios and thus we follow [7] and set the sparsity ratio to $90\%$ and $95\%$. The results are summarized in Table 1. HRBP can still achieve similar performance as these carefully designed unstructured pruning approaches. For example, on CIFAR-10 with ResNet-32, HRBP achieves an accuracy of $90.84\%$ at $95\%$ sparsity, which is $3.65\%$ higher than RC and just $0.17\%$ negligible drop to SNIP [6]. Besides, HRBP usually outperforms

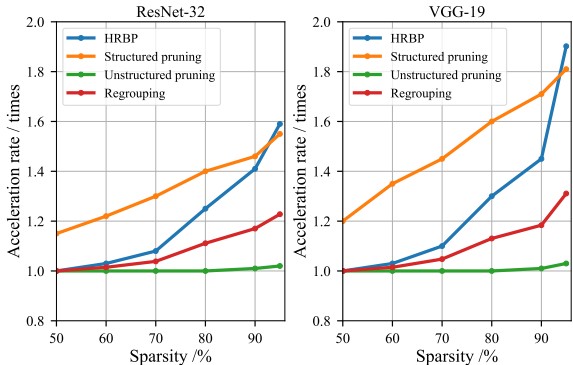

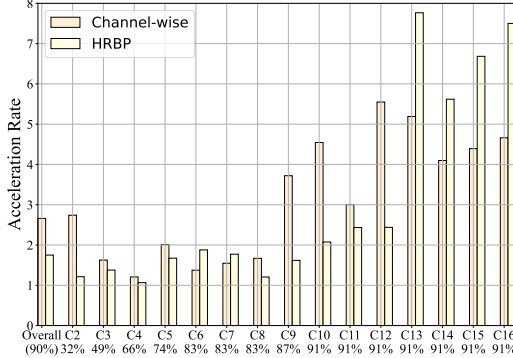

Figure 6: End-to-end training acceleration rate v.s. sparsity ratio of different sparsity schemes.

Figure 7: The layer-wise speedup of VGG-19 at inference time. The number in brackets is the corresponding sparsity.

the vanilla grouping method [21], despite its additional constraints on the shape of dense blocks. Overall, sparse training with HRBP can still match the accuracy of unstructured pruning methods at almost all sparsity levels.

**Acceleration Rate.** We then evaluate the end-to-end training acceleration rate, which includes both forward and backward execution time, and show results in Fig. 6. We observe that HRBP outperforms the vanilla regrouping [21], as the latter only accelerates the forward pass. Remarkably, at large sparsity ratios, HRBP can nearly match the acceleration rate of channel-wise pruning with the benefit of our optimized convolution implementation. In contrast, structured pruning is more effective at lower sparsity ratios, as it directly reduces the weight matrix size. We also investigate the acceleration of each convolutional layer during inference to gain a more detailed understanding. Results are presented in Fig. 7 for VGG-19 with $90\%$ sparsity as an illustration. Similarly, we observe that channel-wise pruning results in slightly better acceleration at shallow layers with small sparsity ratios. However, at deeper layers with larger sparsity ratios, HRBP achieves comparable or even superior results. This finding is consistent with the conclusion of [21] that block-based pruning is more effective for large kernels and high sparsity ratios. Noticeably, `cuDNN` is only optimized for kernel matrices with a multiple of 32 rows [26]. Hence, structured pruning with an arbitrary number of channels does not guarantee better acceleration [21]. In summary, HRBP technique yields comparable accuracy to unstructured pruning while simultaneously accelerating training time on hardware. As a result, our approach presents a superior balance between accuracy and hardware acceleration, especially for high sparsity ratios.

**Results on ImageNet.** To investigate the scalability of our method, we assess SST with HRBP on ImageNet. Following [6, 7], we set the sparse ratio to $60\%$ and $80\%$. As shown in Table 2, HRBP can also achieve similar performance as unstructured pruning methods with well-defined criteria. It provides $1.26\times$ end-to-end training acceleration on hardware when the pruning ratio is $80\%$. Thus, HRBP continues to be efficient for intricate real-world tasks.

Table 2: SST on ImageNet. The dense ResNet-50 has 75.70% top-1 accuracy.

| Sparsity | 60% | | 80% | |
|---|---|---|---|---|
| Accuracy | top-1 | top-5 | top-1 | top-5 |
| SNIP [6] | 73.95 (1.0×) | 91.97 | 69.67 (1.0×) | 89.24 |
| GraSP [7] | 74.02 (1.0×) | 91.86 | 72.06 (1.0×) | 90.82 |
| HRBP | 74.84 (1.17×) | 92.35 | 70.90 (1.26×) | 89.93 |

## 5.3. Dynamic Sparse Training with HRBP

We further explore DST with HRBP, which allows mask updating during training time. We compare DST with HRBP to several unstructured DST methods, including DeepR [39], DSR [37], SET [9], and RigL [10]. We follow all sparse training hyperparameters in [10]. As the speedup is similar to

SST, we only show the accuracy comparison in Table 3 for simplicity. Noticeably, HRBP with block-wise updating can still match the accuracy of several unstructured DST methods. One potential reason is that the mask updating mechanism of HRBP has a smaller mask diversity [24] than the unstructured pruning.

Table 3: Comparison of DST methods on CIFAR-10/100.

| Network | ResNet-32 | | | | VGG-19 | | | |
|---------|-----------|---|---|---|--------|---|---|---|
| Dataset | CIFAR-10 | | CIFAR-100 | | CIFAR-10 | | CIFAR-100 | |
| Sparsity | 90% | 95% | 90% | 95% | 90% | 95% | 90% | 95% |
| Deep-R [39] | 91.62 | 89.84 | 66.78 | 63.90 | 90.81 | 89.59 | 66.83 | 63.46 |
| DSR [37] | 92.97 | 91.61 | 69.63 | 68.20 | 93.75 | 93.86 | 72.31 | 71.98 |
| SET [9] | 92.30 | 90.76 | 69.66 | 67.41 | 92.46 | 91.73 | 72.36 | 69.81 |
| RigL [10] | 92.84±0.13 | 92.02±0.29 | 70.98±0.30 | 68.50±0.15 | 93.15±0.09 | 92.30±0.43 | 71.63±0.28 | 69.13±0.46 |
| HRBP | 92.72±0.23 | 91.25±0.12 | 69.51±0.52 | 66.41±0.30 | 93.07±0.12 | 91.79±0.18 | 70.49±0.19 | 68.04±0.37 |

## 5.4. Ablation Studies

**Block density ratio $s$.** ResNet-32 on CIFAR-10 is used as an example to investigate the impact of the block sparse ratio $s$ in HRBP. Typically, a smaller $s$ introduces more non-zero kernels but inevitably reduces the number of non-zero weights within each kernel. Since most kernels have a shape of $3 \times 3$, we explore $s$ from $\frac{1}{9}$ to $\frac{9}{9}$. As shown in Fig. 8, a smaller $s$ usually leads to slightly better accuracy, particularly for networks with high sparsity, since smaller values of $s$ can retain more non-zero kernels.

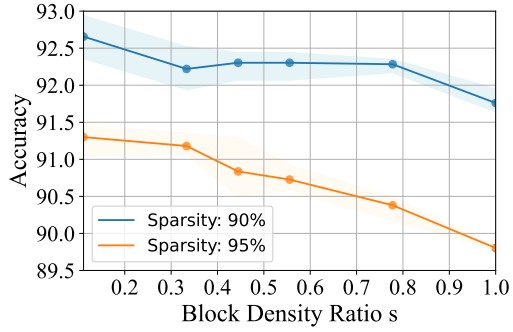

Figure 8: Ablation on kernel dense ratio $s$.

Figure 9: Ablation on mask initialization.

**Mask initialization methods.** We by default apply our method to random unstructured sparse mask with ERK sparse ratio distribution. To investigate the impact of various sparse mask initializations, we apply HRBP to three alternative unstructured masks: 1) random mask with uniform sparse ratio, 2) mask from SNIP [6], and 3) mask from GraSP [7]. The results with different sparsity levels are shown in Fig. 9. Noticeably, with SNIP and GraSP masks, HRBP can also achieve comparable accuracy at the same sparsity level. With uniform sparsity, the accuracy is slightly lower. This is consistent with the conclusion in [10] that ERK sparsity is beneficial. In general, HRBP exhibits robustness to different types of mask initialization.

## 6. Conclusion

This paper reveals that existing regrouping techniques in sparse CNN training are incapable of ensuring backward acceleration. To this end, we propose the HRBP, aiming to accelerate the sparse CNN training at both forward and backward passes. Experimental results suggest that our method can achieve comparable accuracy with unstructured pruning methods at large sparse ratios and brings significant training acceleration on hardware.

## Acknowledgement

This work is partly supported by the National Science Foundation CCF-2312616.

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

# A. Notations

In this section, we summarize all mathematical notations in this paper to help understand the motivation and the operation clearly.

Table 4: Summary of notations in this paper.

| Notation | Range | Definition |
|---|---|---|
| $C_I$ | $\mathbb{N}+$ | Number of input channels |
| $C_O$ | $\mathbb{N}+$ | Number of output channels |
| $K_h$ | $\mathbb{N}+$ | Kernel height |
| $K_w$ | $\mathbb{N}+$ | Kernel width |
| $H_I$ | $\mathbb{N}+$ | Height of input 2D feature map |
| $W_I$ | $\mathbb{N}+$ | Width of input 2D feature map |
| $H_O$ | $\mathbb{N}+$ | Height of output 2D feature map |
| $W_O$ | $\mathbb{N}+$ | Width of output 2D feature map |
| $C_I'$ | $\mathbb{N}+$ | Number of input channels in simplicity form |
| $\mathbf{K}$ | $\mathbb{R}^{C_O \times C_I \times K_h \times K_w}$ | |
| $\mathbf{K}_c$ | $\mathbb{R}^{C_I \times K_h \times K_w}$ | One filter |
| $\mathbf{K}'$ | $\mathbb{R}^{C_I \times C_O \times K_w \times K_h}$ | 2D convolutional weights in backward pass |
| $\mathbf{I}$ | $\mathbb{R}^{C_I \times H_I \times W_I}$ | 2D input feature map |
| $\mathbf{O}$ | $\mathbb{R}^{C_O \times H_O \times W_O}$ | 2D output feature map |
| $\mathbf{dI}$ | $\mathbb{R}^{C_I \times H_I \times W_I}$ | Gradients w.r.t. the 2D input feature map |
| $\mathbf{dO}$ | $\mathbb{R}^{C_O \times H_O \times W_O}$ | Gradients w.r.t. the 2D output feature map |
| $\mathbf{dK}$ | $\mathbb{R}^{C_O \times C_I \times K_h \times K_w}$ | Gradients w.r.t. the kernels |
| $\mathbf{W}$ | $\mathbb{R}^{C_O \times (C_I K_h K_w)}$ | Convolutional weights in GeMM (weights matrix) |
| $\mathbf{W^T}$ | $\mathbb{R}^{(C_I K_h K_w) \times C_O}$ | Transpose of weights matrix |
| $\mathbf{W}'$ | $\mathbb{R}^{C_I \times (C_O K_h K_w)}$ | Convolutional weights of backward pass in GeMM |
| $\mathbf{X}$ | $\mathbb{R}^{(C_I K_h K_w) \times (H_O W_O)}$ | Input feature map in GeMM |
| $\mathbf{Y}$ | $\mathbb{R}^{C_O \times (H_O W_O)}$ | Output feature map in GeMM |
| $\mathbf{dY}$ | $\mathbb{R}^{(C_O K_h K_w) \times (H_I W_I)}$ | Gradients w.r.t. the output feature map in GeMM |
| $\mathbf{dW}$ | $\mathbb{R}^{C_O \times H_O \times W_O}$ | Gradients w.r.t. the Convolutional weights in GeMM |
| $\mathbf{dX}$ | $\mathbb{R}^{C_I \times (H_I W_I)}$ | Gradients w.r.t. the input feature map in GeMM |
| $m$ | $\mathbb{R}^{C_O \times (C_I K_h K_w)}$ | the binary mask of weights matrix $\mathbf{W}$ |
| $m'$ | $\mathbb{R}^{C_O \times C_I}$ | kernel-wise mask |
| $r$ | $[0, 1]$ | dense ratio of one convolutional kernels |
| $t$ | $\mathbb{N}+$ | number of groups (dense blocks) |
| $s$ | $[r, 1] \cap \{i/(K_h K_w)\}_{i=1}^{K_h K_w}$ | dense ratio of each kernel with size $K_h K_w$ |

# B. Applicability to depth-wise CNNs.

Besides the regular convolutional network, we further evaluate our method on MobileNetV2 (dense accuracy 90.56%) to verify its effectiveness on depth-wise CNN, which can be viewed as multiple convolutions with an input channel of $C_I = 1$. In Tab. 5, HRBP still achieves comparable accuracy as SNIP.

Table 5: Static sparse training on MobileNetV2

| Sparsity | 80% | 90% | 95% |
|---|---|---|---|
| SNIP | 90.20% | 88.54% | 83.36% |
| HRBP | 90.54% | 88.20% | 83.50% |

## C. Results on transformers

Recently, the ViT [46] shows that transformers [47] also play an important role in computer vision tasks. The transformer encoder contains a self-attention module and an MLP layer. Given an input sequence $\tilde{\mathbf{X}}$, the self-attention applies three linear transformations to obtain the query $\tilde{\mathbf{Q}} = \tilde{\mathbf{W}}_{\mathbf{q}}\tilde{\mathbf{X}}$, the key $\tilde{\mathbf{K}} = \tilde{\mathbf{W}}_{\mathbf{k}}\tilde{\mathbf{X}}$, and the value $\tilde{\mathbf{V}} = \tilde{\mathbf{W}}_{\mathbf{v}}\tilde{\mathbf{X}}$, respectively. Then the output is obtained by $\tilde{\mathbf{Y}} = \text{Softmax}(\dfrac{\tilde{\mathbf{Q}}\tilde{\mathbf{K}}^{T}}{\sqrt{d}})\tilde{\mathbf{V}}$, where $d$ is the embedding dimension. Thus, all calculations of transformers with learnable weights are linear projections. To this end, the general form $\mathbf{dX} = \mathbf{W}^{T}\mathbf{dY}$ for the gradient w.r.t. the input is suitable for transformers in the backward pass and we can keep the same dense blocks at both forward and backward pass for sparse training of transformers. Although the transformer does not encoder the shape transformation issues like CNN, we further apply our method on DeiT-Tiny [46, 48] to show that the dense-block-based method is also effective in the training of transformers. The dense model of DeiT-Tiny can achieve 72.2% accuracy on ImageNet. With HRBP, we can achieve 72.7% accuracy with 60% sparsity.

## D. Algorithm summary of HRBP

In this section, we summarize the detailed implementation of HRBP in Algorithm 1.

---
**Algorithm 1** HRBP
---
**Input:** unstructured mask $m$, density ratio of mask $r$, number of clusters $t$, density ratio of kernel $s$, number of input channel $C_I$, the number of output channel $C_O$, the width of the kernel $K_w$, the height of the kernel $K_h$, and the minimum number of rows of each group $B_1$
**Output:** Dense groups
Obtain kernel-wise mask $m'$ based on the number of non-zero cells within each kernel of $m$;
Divide the rows in $m'$ into $t$ equal-shape groups $\{g_1, g_2, ..., g_t\}$ with hypergraph partitioning;
**for** $g_i \in \{g_1, g_2, ..., g_t\}$ **do**
  **if** $g_i$ has no less than $B_1$ rows **then**
    Sort columns of $g_i$ from high to low based on the number of non-zero cells of each column;
    Select the $\frac{r}{s}C_I$ columns with the most number of non-zero;
    Extract the corresponding kernels $\mathcal{K}$ based on selected columns in $g_i$;
    Count the non-zero weights of each cell based on all kernels in $\mathcal{K}$;
    Select $sK_hK_w$ cells with the most number of non-zero weights as the pattern $P$;
    Apply pattern $P$ to all kernels $\mathcal{K}$ and output them as a dense group;
  **end if**
**end for**
---

## E. HRBP Hardware Implementation

We implement our method with CUDA and measure the acceleration rate on GPU. In detail, we rewrite the `nn.Conv2d` module of Pytorch with CUDA to run CNN with sparse weights. In CUDA programming, threads serve as the basic programmable unit and allow GPU programmers to leverage massive numbers of CUDA cores. These threads are grouped at different levels such as warp, block, and grid. HRBP decomposes the original convolution computation into several computations corresponding to dense blocks, and these computations will be performed in parallel by threads. As the extracted dense blocks from HRBP have the same shape, in our kernel design, a thread block only processes the output channels of the same group, which enables us to have better reuse of the input data (e.g., the input data can be loaded to shared memory which can be accessed by all the threads inside a thread block). Inside a thread block, each thread is responsible for one portion of the output on $(X, Y)$ dimension which further helps us to achieve good data reuse of the kernel weight (kernel weight can also be put into shared memory). We also applied tiling across both height and width dimension of the input channel to increase the number of launched thread blocks, which further increases the parallelism level. The tiling size plays a significant role of the

performance. A small tiling size can increase the parallelism level, while a big tiling size will provide better reuse of the weight data. Thus, finding a good tile size for a given problem is non-trivial. A simple but powerful method is brute force search, it guarantees to find out the best tiling size for specific hardware. In our design, input data are put into shared memory and can be accessed by all thread block to improve the data reuse. In the meantime, the cost by using brute force search is very low because most of the undesired tiling size candidates are discarded in order to satisfy the usage requirement of the shared memory.

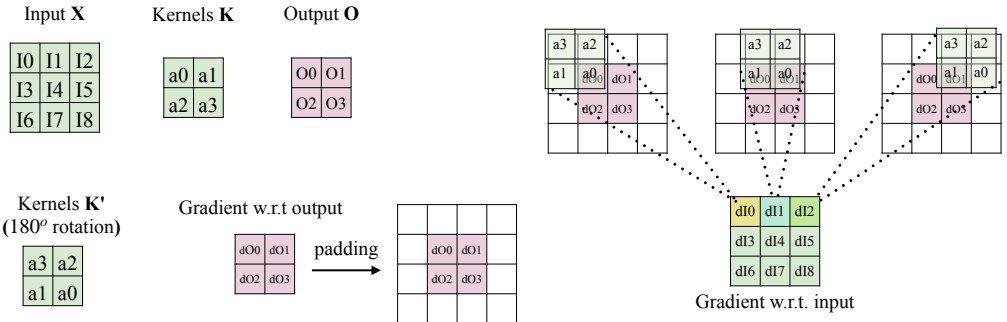

Figure 10: Illustration of the full convolution between the rotated kernels $\mathbf{K}'$. For simplicity, we use $dO$ to represent $\frac{\partial L}{\partial O}$.

## F. Mathematical Analysis of CNN backward propagation

In this section, we give a detailed mathematical analysis of the CNN backward pass to show the reason for the transformation in the calculation of gradient w.r.t. the input. For simplicity, we assume that the number of input channels $C_I$ is 1, the number of output channels $C_O$ is 1, the stride is 1, and there is no padding operation in the forward pass. Thus, given the input $X \in \mathbb{R}^{H_I \times W_I}$ and the filter $K \in \mathbb{R}^{K_h \times K_w}$, in the forward pass of CNN, the output $O \in \mathbb{R}^{H_O \times W_O}$ is calculated by:

$$\mathbf{O}_{i,j} = \sum_{m=1}^{K_h} \sum_{n=1}^{K_w} K_{m,n} X_{i+m,j+n}, \tag{1}$$

where $1 \le i \le H_O$ and $1 \le j \le W_O$. As there is no padding operation and the stride is 1, we have $H_O = H_I - K_h + 1$ and $W_O = W_I - K_w + 1$. In the backward pass, given the gradients of the output $\frac{\partial L}{\partial O}$, based on the chain rule, the gradient w.r.t. each input pixel is calculated by:

$$\frac{\partial L}{\partial X_{a,b}} = \sum_{i=1}^{H_O} \sum_{j=1}^{W_O} \frac{\partial L}{\partial O_{i,j}} \frac{\partial O_{i,j}}{\partial X_{a,b}}, \tag{2}$$

where $1 \le a \le H_I$ and $1 \le b \le W_I$, and $\frac{\partial O_{i,j}}{\partial X_{a,b}}$ can be obtained by taking the difference on both sides of Equation 1. Thus, when input pixel $X_{a,b}$ contributes to the output pixel $O_{i,j}$, $\frac{\partial L}{\partial O_{i,j}}$ contributes to the gradient of $\frac{\partial L}{\partial X_{a,b}}$. In practice, Equation 2 can be represented as a full convolution between a 180-degree rotated filter $\mathbf{K}'$ and the gradient on the output.

Considering a specific example where the input has shape $H_I = W_I = 3$, and the kernel has size $K_h = K_w = 2$, as shown in Figure 10. For simplicity, we set $CI = C_O = 1$. In the forward, based on Equation 1, we have:

$$O_0 = I_0 * a_0 + I_1 * a_1 + I_3 * a_2 + I_4 * a3$$

$$O_1 = I_1 * a_0 + I_2 * a_1 + I_4 * a_2 + I_5 * a3$$
$$O_2 = I_3 * a_0 + I_4 * a_1 + I_6 * a_2 + I_7 * a3$$
$$O_3 = I_4 * a_0 + I_5 * a_1 + I_7 * a_2 + I_8 * a3$$

Take $O_0$ as an example, the gradient w.r.t. $I_0$, $I_1$, $I_3$, and $I_4$ are $\frac{\partial O_0}{\partial I_0} = a_0$, $\frac{\partial O_0}{\partial I_1} = a_1$, $\frac{\partial O_0}{\partial I_3} = a_2$, and $\frac{\partial O_0}{\partial I_3} = a_3$ respectively. Based on Equation 2, we can obtain the gradient of each input pixel:

$$\frac{\partial L}{\partial I_0} = \frac{\partial L}{\partial O_0} * a_0$$
$$\frac{\partial L}{\partial I_1} = \frac{\partial L}{\partial O_0} * a_1 + \frac{\partial L}{\partial O_1} * a_0$$
$$\frac{\partial L}{\partial I_2} = \frac{\partial L}{\partial O_1} * a_1$$
$$\frac{\partial L}{\partial I_3} = \frac{\partial L}{\partial O_0} * a_2 + \frac{\partial L}{\partial O_2} * a_0$$
$$\frac{\partial L}{\partial I_4} = \frac{\partial L}{\partial O_0} * a_3 + \frac{\partial L}{\partial O_1} * a_2 + \frac{\partial L}{\partial O_2} * a_1 + \frac{\partial L}{\partial O_3} * a_0$$
$$\frac{\partial L}{\partial I_5} = \frac{\partial L}{\partial O_1} * a_3 + \frac{\partial L}{\partial O_3} * a_1$$
$$\frac{\partial L}{\partial I_6} = \frac{\partial L}{\partial O_2} * a_2$$
$$\frac{\partial L}{\partial I_7} = \frac{\partial L}{\partial O_2} * a_3 + \frac{\partial L}{\partial O_3} * a_2$$
$$\frac{\partial L}{\partial I_8} = \frac{\partial L}{\partial O_3} * a_3$$

As in Figure 10, we can perform a convolution operation between the 180-degree rotated kernels and the gradient w.r.t output $\frac{\partial L}{\partial O}$ with zero padding. Note that, zero padding on the output is necessary as we need to ensure the product of this full convolution has the same shape as the input.

## G. Forward and Backward pass in `NCHW` and `NHWC` layout

`NCHW` and `NHWC` data layout formats are two common types of cuDNN tensors arrangement in memory [26]. These two layouts produce the same shape matrix and the same outputs. The only difference is the order of element in the flattened tensor, as shown below:

`NCHW` **layout** As shown in Fig. 11(a), in forward pass, the flattened tensor $\mathbf{W}$ begins with the first input channel (green color), the elements are arranged contiguously in row-major order (*i.e.*, $a0, a1, a2, a3$ with green color for the kernel $\mathbf{K}$ in forward). Then, it continues with second (orange color) and subsequent channels until the elements of all the channels are laid out.

`NHWC` **layout** As shown in Fig. 11(c), in the forward pass, the flattened tensor $\mathbf{W}$ begins with the first element of the first input channel (*i.e.*, $a0$ with green color for $\mathbf{K}$), then proceed to the first element of the second input channel (*i.e.*, $a0$ with orange color for $\mathbf{K}$), and so on, until the first elements of all the $C$ channels are laid out. Next, select the second element of the first input channel (*i.e.*, $a1$ with green color for $\mathbf{K}$), then proceed to the second element of the second input channel (*i.e.*, $a1$ with orange color for $\mathbf{K}$), and so on, until the second element of all the channels is laid out.

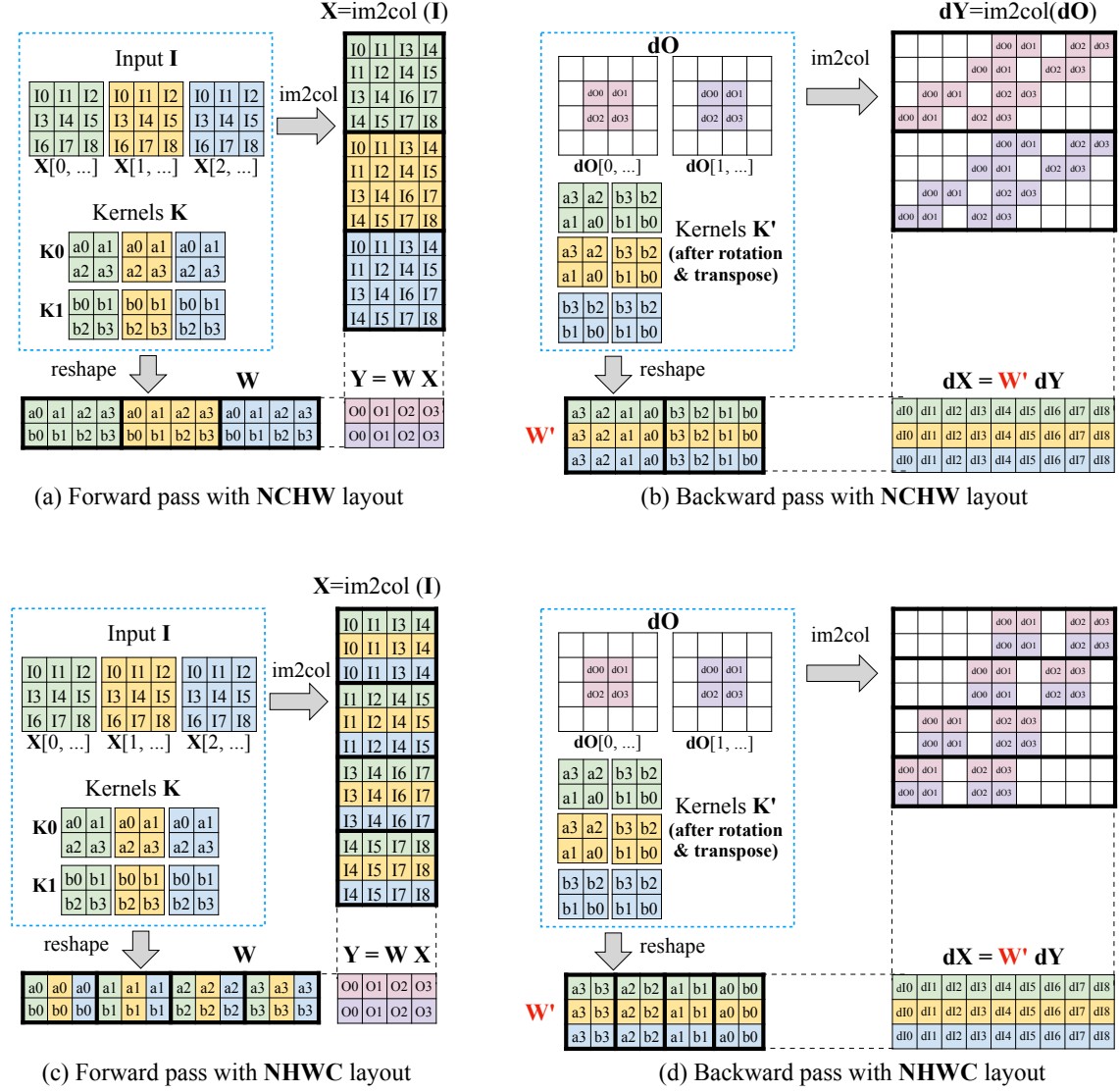

Figure 11: Illustration of GeMM with `NCHW` and `NHWC` format. Different colors represent different channels.

## H. N:M Transposable Mask in the Backward of CNN

As suggested in [20], the N:M sparsity The N:M transposable mask [24] guarantees that both the weights matrix $\mathbf{W}$ and its transpose $\mathbf{W}^T$ follow the same sparsity pattern. This design works well for linear layers, whose forward pass is calculated with $Y = WX$ and backward pass is calculated with $dX = W^T dY$. However, the implementation of CNN with GEMM is different. Specifically, we have $\mathbf{W} \in \mathbb{R}^{C_O \times (C_I K_h K_w)}$ for the weights matrix of CNN. In the backward pass, CNN flips each kernel first vertically and then horizontally and performs the kernel-wise transpose. Thus, $\mathbf{W}^T \in \mathbb{R}^{(C_I K_h K_w) \times C_O}$ is different from the matrix $\mathbf{W}' \in \mathbb{R}^{C_I \times (C_O K_h K_w)}$ in the backward pass of CNN in most cases. Nevertheless, this N:M transposable mask can still accelerate the CNN backward under some conditions.

**Layout.** The first requirement is the layout of the weights in the memory. For example, suppose $N = 2$, $M = 4$, the size of the input channel is $C_I = 4$ and the size of the output channel is $C_O = 4$. Thus, we can divide the kernel matrix $\mathbf{W} \in \mathbb{R}^{4 \times (4 \times 3 \times 3)}$ into several $4 \times 4$ blocks. As shown in Fig. 12(a)(b), when we store kernels with `NHWC` format, we collect $a0$ with green color, $a0$ with

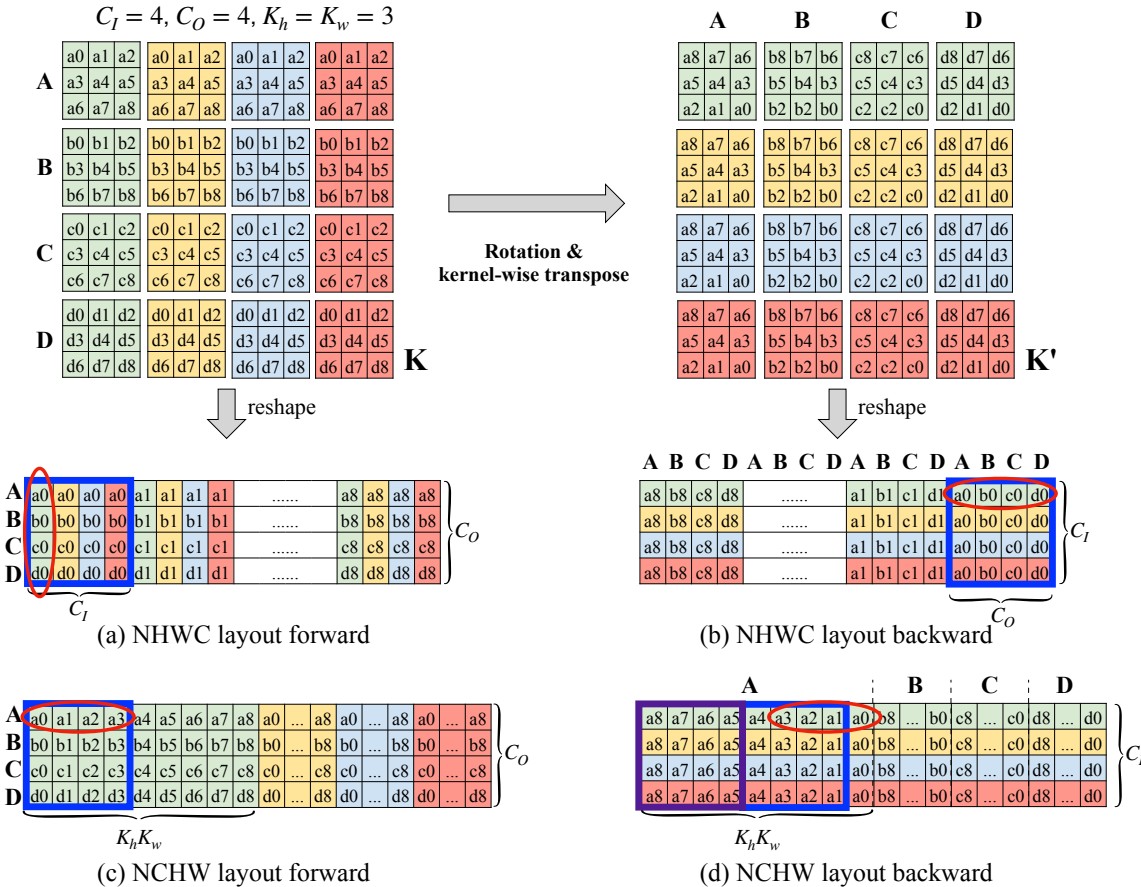

Figure 12: Comparison of `NHWC` and `NCHW` layouts of N:M mask in both forward and backward pass.

orange color, $a0$ with blue color, and $a0$ with red color into a row of a block (*i.e.*, the blue box) for the forward pass, and this row meets the N:M constraint. Based on the N:M transposable mask, the column of $a0$, $b0$, $c0$, and $d0$ with green color also satisfies the N:M constraint. In the backward pass, this blue block can still be kept. Meanwhile, $a0$, $b0$, $c0$, and $d0$ with green color is collected as a row. Thus, the same sparse pattern can be maintained in the backward with N:M transposable design. When the weights is stored with `NCHW` format, as shown in Fig. 12(c)(d), the green cell $a0$, $a1$, $a2$, and $a3$ form a row of the blue block. Again, the column of $a0$, $b0$, $c0$, and $d0$ with green color also satisfies the N:M constraint. However, in the backward pass, the same block cannot be kept. Thus, the sparse pattern may not meet the N:M constraint. For example, in forward pass, green cells $a2$ and $a3$ are kept, and the green cells $a4$ and $a7$ are kept. In the backward, the green cell $a1$, $a2$, $a3$, and $a4$ form a new vector with the size $M = 4$, but this vector has three cells $a2$, $a3$, and $a4$ with mask "1". To this end, additional operations such as re-indexing and regrouping are required to make the weight matrix in the backward meet the N:M constraint. In summary, the N:M transposable mask can accelerate the CNN backward pass with `NHWC` layout GEMM implementation, but it cannot be directly applied to the `NCHW` layout.

**Shape.** Moreover, even if the weights are arranged in `NHWC` layout, we show some cases that the N:M transposable mask may not bring the acceleration on backward as well. Considering the example with $C_O = 3$ and $M = 2$, as shown in Fig. 13, we collect $a0$ with green color, $a0$ with orange color into a row of a $2 \times 2$ block (*i.e.*, the blue box) for the forward pass, and this row meets the N:M constraint. Based on the N:M transposable mask, the column of $a0$, $b0$ with green color also satisfies the N:M constraint. However, in the backward, the $a0$ is grouped with $c1$, and the $b0$ is grouped

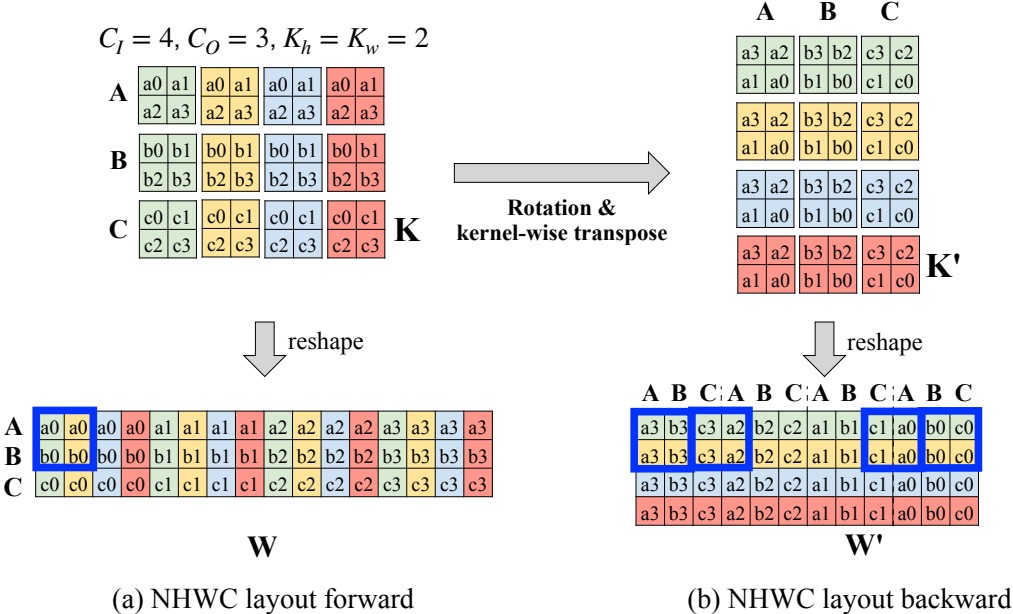

(a) NHWC layout forward

(b) NHWC layout backward

Figure 13: Example of N:M transposable mask when $C_O \mod M \neq 0$.

with $c0$. Thus, the same sparsity pattern cannot be guaranteed. To this end, the output channel $C_O$ should be divisible by $M$, *i.e.*, $C_O \mod M = 0$.

