# OpenReview forum: "HRBP: Hardware-friendly Regrouping towards Block-based Pruning for Sparse CNN Training"
_CPAL.cc/2024/Conference — CPAL 2024 (Proceedings Track) Oral_

### Official Review · Reviewer_gs6o · 2023-10-03
**Strong motivation and clear methodology, but the evaluation results are incomplete.**

**Rating:** 5
**Confidence:** 3

**Review:**

### Strength:
* The paper clearly proposes a significant concern: the existing fine-grained structured pruning algorithm struggles to accelerate backpropagation when used with specific scenarios, such as CNN with GEMM. Section 3 delves deeply into this issue and presents a compelling argument.
* The method proposed focuses on regrouping using a kernel-wise mask and integrates pattern finding. This seems to be a sound approach to tackling the identified issues.

### Weakness:
* The presented results are incomplete and lack organization. In Table 1, for instance, the speedup values (which should be indicated in brackets as said in the caption) are missing. To grasp the benefits of the proposed method, readers have to cross-reference both Figure 5 and Figure 6, as these figures jointly demonstrate that the method offers improved speed without compromising performance. Notably, Figure 6 omits the ResNet-56 configuration, making it inconsistent with Figure 5.

* The paper does not provide sufficient details about its evaluation. The procedure by which training acceleration was gauged remains unclear, and the experimental setup is not adequately defined.

---

### Official Review · Reviewer_Turq · 2023-10-04

**Rating:** 7
**Confidence:** 4

**Review:**

This paper investigates practical hardware acceleration upon unstructured sparse training. Specifically, the authors offered a comprehensive analysis of the acceleration bottlenecks encountered by previous methods during backpropagation and then proposed a kernel-wise mask for grouping unstructured sparse weights, achieving effective acceleration during backpropagation. The efficacy of the proposed method is demonstrated through experiments on CIFAR and ImageNet. The reviewer acknowledges the contribution of this paper and also makes some suggestions as follows:

1. The author's analysis of GEMM is very thorough, which is highly appreciated. Nonetheless, I would like to raise two points. First, NVIDIA's sparse tensor core is implemented based on NHWC[1], so the T-mask itself devised for N:M pattern is acceptable. Moreover, the N:M sparsity of the network will not be applied to the situation where C0=3, which has been given in Nvidia's documentation, because it is inherently unsuitable. Secondly, I would like to point out that the authors' mentioned inapplicability of T-mask under NCHW situation can be overcome using a different, recently proposed BI-Mask[2]. Including a discussion around this might make the paper more detailed.

2. The kernel-wise mask is very innovative and easily understood, and its design thought that integrates hardware design is quite reasonable. It would be a significant contribution to the field if the author could open-source the related acceleration code.

3. On ImageNet, the authors mostly compared PAI methods. It would be better to compare with DST methods because accelerating training on ImageNet is more critical than smaller datasets like CIFAR. Even if the performance is not as good as DST, it can still show the user a trade-off between training acceleration and performance.

[1] Nvidia a100 tensor core gpu architecture. https://www.nvidia.com/content/dam/en-zz/Solutions/Data-Center/nvidia-ampere-architecture-whitepaper.pdf, 2020
[2] Bi-directional Masks for Efficient N: M Sparse Training. In ICML, 2023.

---

### Official Review · Reviewer_2ePh · 2023-10-11

**Rating:** 6
**Confidence:** 3

**Review:**

## Summary:
The paper introduces HRBP, a block-wise pruning method for CNNs. HRBP maintains block-wise sparsity in both the forward and backward passes of CNN training, leading to improved efficiency.

## Strengths:

The idea of preserving block-wise sparsity in the backward pass is well-motivated.
Empirical results consistently demonstrate HRBP's performance improvements in static and dynamic training.

## Weaknesses & questions from the reviewer:

- To me, the technical exposition in the paper is quite challenging to follow. To clarify the core idea of HRBP, would it be possible to include a simple walk-through example, such as applying HRBP to a 1D convolution? This would demonstrate how HRBP maintains sparsity in both forward and backward passes in circulant matrices.

-  In comparison to traditional block-pruning methods like [21], HRBP preserves sparsity in the backward pass. This is expected to enhance gradient computation efficiency, resulting in shorter run times. However, the experimental results also indicate that HRBP improves training accuracy compared to baseline methods. It would be insightful if the authors could provide some intuition behind this observation.

---

### Official Review · Reviewer_UHbo · 2023-10-15
**Hardware friendly pruning for accelerating forward AND backward passes**

**Rating:** 8
**Confidence:** 4

**Review:**

This paper presents a pruning methodology which yields speed-ups of sparse neural networks in both forward and backward passes. The key intuition comes from how forward / backward passes are calculated in GEMM. As I understand it, the method prunes weights at the level of a single weight kernel, as indexed by input AND output channel. This guarantees that sparsity can be exploited to accelerate both forward and backward passes. This idea seems quite intuitive and simple. The results section shows that this pruning approach yields speed-ups on par with structured (channel-wise) pruning while exhibiting the accuracy of unstructured pruning. Authors also show results for dynamic sparsity training, which is an added plus. I rate this paper quite highly, as it shows real speed-ups on real hardware used by a large number of practitioners / researchers today.

---

### Meta-Review · Area_Chair_BUUv · 2023-11-14

**Recommendation:** Accept (Poster)
**Confidence:** 5

**Metareview:**

Sparse DNNs are a prominent topic both in practice and in theory of deep learning but there are currently few off-the-shelf, simple ways to exploit weight sparsity. There is little low-level work which demonstrates important improvements with real data on real hardware.
Here we have a refreshingly sober paper which addresses this issue and designs a simple, creative solution which is demonstrated to work well in practice, with speed-ups on par with structured pruning methods and accuracy on par with unstructured methods. This could have substantial impact in practice. All reviewers agree that this is a solid contribution and recommend acceptance (either in score or in comments).

---

### Meta-Review · Program_Chairs · 2023-11-19

**Recommendation:** Accept (Oral)
**Confidence:** 5

**Metareview:**

Overall, the reviewers and AC agreed that the paper makes a significant contribution to the field of sparse training. The paper proposes a novel hardware-friendly block-wise pruning method for sparse CNN training. The method preserves block-wise sparsity in both the forward and backward passes of CNN training, leading to improved efficiency. The paper is well-written and the experimental results are convincing.

The action PC chair for this paper is Gintare Karolina Dziugaite, who made the decision after carefully reading the paper as well as the comments by all reviewers and AC. The decision is agreed by all PC chairs.

---

### Decision · Program_Chairs · 2023-11-20

**Decision:**

Accept (Oral)

**Comment:**

Overall, the reviewers and AC agreed that the paper makes a significant contribution to the field of sparse training. The paper proposes a novel hardware-friendly block-wise pruning method for sparse CNN training. The method preserves block-wise sparsity in both the forward and backward passes of CNN training, leading to improved efficiency. The paper is well-written and the experimental results are convincing.

The action PC chair for this paper is Gintare Karolina Dziugaite, who made the decision after carefully reading the paper as well as the comments by all reviewers and AC. The decision is agreed by all PC chairs.